# Natural Plant Extracts and Compounds for Rheumatoid Arthritis Therapy

**DOI:** 10.3390/medicina57030266

**Published:** 2021-03-15

**Authors:** Xiangyu Zhao, Young-Rok Kim, Yunhui Min, Yaping Zhao, Kyoungtag Do, Young-Ok Son

**Affiliations:** 1Interdisciplinary Graduate Program in Advanced Convergence Technology and Science, Jeju National University, Jeju 63243, Korea; zhaoxiangyu@jejunu.ac.kr (X.Z.); reinise4011@jejunu.ac.kr (Y.M.); 2Department of Animal Biotechnology, Faculty of Biotechnology, Jeju National University, Jeju 63243, Korea; Kcia7509@hanmail.net; 3School of Chemistry and Chemical Engineering, Frontiers Science Center for Transformative Molecules, Shanghai Jiao Tong University, Shanghai 200240, China; ypzhao@sjtu.edu.cn; 4Bio-Health Materials Core-Facility Center, Jeju National University, Jeju 63243, Korea; 5Practical Translational Research Center, Jeju National University, Jeju 63243, Korea

**Keywords:** natural plant, rheumatoid arthritis, therapy

## Abstract

Natural plant extracts and compounds (NPECs), which originate from herbs or plants, have been used in the clinical treatment of rheumatoid arthritis (RA) for many years. Over the years, many scientists have carried out a series of studies on the treatment of RA by NPEC. They found a high quantity of active NPECs with broad application prospects. In view of various complex functions of these NPECs, exploring their potential as medicines for RA treatment will be beneficial for RA patients. Thus, to help advance the development of high-quality NPECs for RA, we herein aimed to review the research progress of NPECs in the treatment of RA in recent years. Our findings showed that, from the pharmacological perspective, natural plant extracts or mixed herbal compounds effectively regulate the immune system to alleviate RA by inhibiting pro-inflammatory cytokines. Further, individualized medication can be applied according to each patient’s physical condition. However, the pathogenesis of RA and its immune mechanism has not been fully understood and requires further studies.

## 1. Introduction

Rheumatoid arthritis (RA) is a chronic systemic inflammatory autoimmune disease of the joint [1]. Systemic autoimmune disease affects many non-joint organs, such as the skin, eyes, lungs, heart, kidneys, salivary glands, nerve tissue, bone marrow, and blood vessels [2]. However, RA mainly attacks the joints. This common immunological disease hinders the activity and function of the joints, affecting the health and quality of life of afflicted patients. According to epidemiological statistics, the incidence of RA is approximately 1% of the worldwide population; that is, nearly 700 million people worldwide have RA, and more than 80% of the patients are women [3]. 

Inflammatory cytokines play an essential role in the occurrence and development of RA. For example, TNF-α is a pro-inflammatory factor that causes the activation and aggregation of the cell inflammasome. This induces the release of other inflammatory mediators and aggravates the inflammatory response [4]. IL-2 is secreted by activated Th1 cells, which helps lymphocyte and T cell proliferation as well as inducing local inflammatory response [5]. IL-13 is produced by Th2 cells and promotes eosinophil production [6]. Recent studies have shown that antigen-activated CD4+ T cells, monocytes, macrophages, and synovial fibroblasts can produce many inflammatory factors, including TNF-α, IL-1, and IL-6, leading to the secretion of metalloproteinases by chondrocytes, fibroblasts, and osteoclasts [4]. Subsequently, the erosion of bone and cartilage causes the gradual destruction and functional loss of the joints [7]. 

RA patients are required to change their lifestyle [8]. Medication for RA mainly involves non-steroidal anti-inflammatory drugs, anti-rheumatism drugs, and glucocorticoid drugs [9]. In recent years, preclinical trials have proved that natural plant extracts and compounds (NPECs) can significantly alleviate RA [10]. Considering that NPEC medicines for the treatment of RA present various complex functions, exploring the potential of NPECs as medicines for RA treatment will be beneficial for RA patients. Therefore, we herein review the recent research progress of NPECs as a treatment of RA. This article will help advance the development of high-quality NPECs for RA.

## 2. Natural Plant Extract (NPE)

### 2.1. Cinnamomum cassia Presl

*Cinnamomum cassia* Presl, also known as cassia or cinnamon, is a tropical aromatic evergreen tree of the Lauraceae family, commonly used in traditional Oriental medicine. More than 160 chemicals have been identified from *C. cassia.* The main constituents are terpenoids, phenylpropanoids, and glycosides [11]. The main components of *C. cassia* have a wide range of pharmacological effects, including anti-platelet aggregation, antithrombotic, pro-angiogenesis, vasodilating, and microcirculation-improving effects [12]. In addition, *C. cassia* has antitumor, anti-inflammatory, analgesic, antibacterial, antiviral, cardiovascular-protective, cytoprotective, neuroprotective, immunomodulatory, and anti-tyrosinase activities [11,12].

Terpenoids, phenylpropanoids, and glycosides in *C. cassia* have immunomodulatory ability and can reduce the levels of inflammatory mediators, such as interleukin (IL)-1β, transforming growth factor-α (TGF-α), and prostaglandin E2 (PGE2) in the synovial fluid [13]. Western blotting analysis revealed that the expression of cyclooxygenase (COX)-2 and inducible nitric oxide synthase (iNOS) was significantly reduced by *C. cassia*, indicating the suppression of immune responses and alleviation of joint inflammation [13].

In a recent study, cinnamaldehyde (CA) in *C. cassia* extract was shown to exert anti-inflammatory effects against RA. The therapeutic effects of CA were revealed in in vitro experiments using activated macrophages (Raw246.7 cells) and in a rat model of adjuvant arthritis (AA) in vivo [14]. CA is an α,β-unsaturated aromatic aldehyde that can be used as a flavoring agent (Figure 1). It is the principal flavor component of cinnamon oil. The researchers concluded that CA is a potential therapeutic compound that can inhibit RA progression by suppressing IL-1β by modulating the succinate/HIF-1α axis and inhibiting NLRP3 [14]. Moreover, CA significantly reduced synovial inflammation in AA rats and in the peripheral blood mononuclear cells of RA patients by inhibiting the expression of pro-inflammatory cytokines (IL-1β, TNF-α, and IL-6) [15,16]. The binding of CA on the residues of TNF-α and IL-6 was described using a molecular docking experiment [16].

Further studies have found that CA inhibited the activity of HIF-1α by inhibiting the accumulation of succinate in the cytoplasm [15]. To the best of our knowledge, the reduction of HIF-1α nuclear location slows down the production of IL-1β through inhibition of the NLRP3 assembly of inflammasome [14]. In addition, CA inhibited the expression of the succinate receptor GPR91, thereby inhibiting the activation of HIF-1α [14,15].

### 2.2. Ligusticum chuanxiong Hort

*Ligusticum chuanxiong* Hort (Umbelliferae), also called Chuanxiong Rhizoma, is a medicinal herb that has been extensively applied to treat various diseases. The main components of *L. chuanxiong* are ligustilide, 3-butyrolactone, and cypressene [17]. It also contains ferulic acid, tetra-methylpyrazine (ligustrazine or chuanxiongzine), palmitic acid, carotene, and β-sitosterol. Many biomedical and clinical studies have shown the antioxidant, neuroprotective, anti-fibrosis, antinociceptive, anti-inflammatory, and antineoplastic activities of *L. chuanxiong* [18,19,20].

Chuan-Xian Mu et al. (2014) showed that ligustrazine can significantly inhibit swelling in a rat model of collagen-induced arthritis (CIA) [21]. Serum IL-1 and IL-6 levels were decreased, whereas serum IL-2 levels were increased by treatment with ligustrazine [21]. The results suggested that ligustrazine inhibited RA by elevating the levels of anti-inflammatory cytokines and maintaining the balance of the inflammatory cytokine network [21]. More importantly, treatment using a combination of leflunomide, a disease-modifying antirheumatic drug (DMARD) for RA, and ligustrazine attenuated bone erosion in RA patients [22].

### 2.3. Aconitum kusnezoffii Reichb.

The dry root of *Aconitum kusnezoffii* Reichb. (caowu) has been used to treat RA and joint pain for many years owing to its anti-inflammatory properties. Pharmacological studies have shown that diterpenoid alkaloids, including aconitine, mesaconitine, hypaconitine, neoline, talatizamine beiwutine, and deoxy-aconitine, are responsible for the main bioactive effects of *A. kusnezoffii* [23].

Recent research showed that benzoylaconitine (BAC) from the root of *A. kusnezoffii* has potent anti-inflammatory effects (it inhibits the production of IL-6 and IL-8 in human synovial cells) [24,25]. Encapsulated mPEG-PLGA NPs (NP/BAC) with improved bioavailability provide a promising RA therapy strategy, exhibiting high cytocompatibility for activated macrophages. NP/BAC significantly inhibited the secretion of the pro-inflammatory cytokines TNF-α and IL-1β (60–70%) by inhibiting the NF-κB signaling pathway [25,26]. NP/BAC also attenuated ear (69.8%) and paw (87.1%) swelling in an in vivo CIA model [25,26].

Pharmacological researchers believe that Aconiti Kusnezoffii Radix (caowu) exerts an inhibitory effect on the immune response and an antioxidant effect. Caowu reduces painful obstruction syndrome, relieves pain and is widely used in RA treatment to alleviate arthralgia and pain [27].

### 2.4. Tripterygium wilfordii Hook F

Tripterygium wilfordii Hook F (TWHF) has a long history of use for ameliorating RA symptoms. TWHF has various pharmacological activities, such as antitumor, anti-inflammatory, and immune system-regulatory activities [28].

The chemical composition of TWHF is complex, and many biologically active substances have been isolated. The identified compounds include sesquiterpenes, diterpenes (triptolide, tripdiolide, and triptonide), triterpenes (celastrol, pristimerin, and wilforlide A), lignans, glycosides, and alkaloids [28,29,30]. Triptolide and celastrol are considered the representative active components of TWHF, with higher percentages of content and clinical application prospects.

In a recent clinical follow-up study by Zhou et al., patients with RA were treated with TWHF for two consecutive years [31]. Clinical indexes and radiographic data were collected for 2 years and compared using intent-to-treat and per-protocol analyses. A total of 109 patients completed the test during the two-year therapy period. The research concluded that TWHF monotherapy was not inferior to methotrexate monotherapy in patients with RA [31]. 

Triptolide treatment inhibited serum inflammatory cytokine levels in rats with CIA-induced RA [32]. TWHF significantly inhibited increases in IL-1β and TNF-α levels and significantly decreased the levels of the pro-inflammatory cytokines IL-17 and IL-8 [33]. Moreover, the expression of vascular endothelial growth factor (VEGF) and toll-like receptor 2 (Tie2) in rat synovial cells was downregulated by triptolide [34]. The expression of angiogenin-1 (Ang-1) and Ang-2 was also markedly decreased by triptolide in CIA-induced RA rats [34]. The results showed that triptolide improved the severity of CIA-induced RA by inhibiting the RANKL-mediated ERK/AKT signaling pathway in rat synovial cells [34]. Triptolide regulates the proportion of CD4^+^ and CD8^+^ populations and suppresses T and B lymphocytes [35]. Furthermore, triptolide attenuates the expression of TCR receptors in rats with CIA [36]. In addition, celastrol treatment decreased Th17 population, but increased Treg population in arthritic joints [37]. These results suggested that triptolide and celastrol can suppress the immunological function of RA rat models.

### 2.5. Curcumae Longae Rhizoma

Curcumae Longae Rhizoma (CLR) is a traditional herbal medicine that has been used for many years, and mainly contains volatile oil and phenolic substances [38]. Several components have been identified in the volatile oils of CLR, mostly terpenoids, curcumone, and gingerene. The main phenolic component of CLR is curcumin [38,39]. CLR also contains a new turmeric ketone, 3-sitosterol, 3-sitosterol-3-O-carotene, and turmeric polysaccharides [39,40]. Pharmacological studies have shown that CLR protects the liver and exerts choleretic, antibacterial, anti-inflammatory, antitumor, blood lipid-lowering, and anti-pathogenic microorganism effects [41]. It also protects the digestive system, enhances immune function, and improves coronary blood flow in the heart [40,41,42,43,44,45].

β-Elemene is a natural compound extracted from CLR. Elemenes, which include α-, β-, γ-, and δ-elemene, are structural isomers of each other and are classified as sesquiterpenes (Figure 2). β-Elemene significantly inhibited the viability and promoted the apoptosis of human RA fibroblast-like synoviocytes in a concentration-dependent manner [46]. β-Elemene significantly decreased mitochondrial membrane potential and cytochrome c accumulation in the cytosol, as well as increased caspase-9 and caspase-3 activities [47]. All of these activities are related to apoptosis signaling, and this phenomenon was reversed by pretreatment with the p38 inhibitor SB203580 or the reactive oxygen species (ROS) inhibitor N-acetyl-l-cysteine [47]. β-Elemene effectively induces mitochondrial apoptosis in fibroblast-like synoviocytes, and this effect is mediated via induction of ROS formation and p38 mitogen-activated protein kinase (MAPK) activation [48]. This result suggested that β-elemene has therapeutic potential against RA [48].

Curcumin incorporates a seven-carbon linker and three major functional groups: an α,β-unsaturated β-diketone moiety and an aromatic O-methoxy-phenolic group (Figure 3). Curcumin from CLR can reduce Complete Freund’s Adjuvant (CFA)-induced glial cell activation and inflammatory mediator levels IL-1β, monocyte chemotactic protein-1 (MCP-1), and monocyte inflammatory protein in the spinal cord-1 (MIP-1α)] [49,50]. Curcumin also reduces the production of IL-1β, TNF-α, MCP-1, and MIP-1α in lipopolysaccharide (LPS)-stimulated astrocytes and microglia cells [49,51]. Curcumin alleviates arthritis pain by inhibiting the activation of glia and the production of inflammatory mediators in the spinal cord in a rat model of mono-arthritis and thus has a potential for treating arthritis pain [51]. RA patients who received either a low (250 mg) or high (500 mg) dose of curcumin (twice daily for 90 days) showed significantly improved clinical symptoms via the American College of Rheumatology response, visual analog scale, C-reactive protein, Disease Activity Score 28, erythrocyte sedimentation rate, and rheumatoid factor values, compared with those who received placebo, without any side effects [52]. The weight of the immune organ of rat RA models indicated the immunological inhibition effects of curcumin [53]. These findings suggest that curcumin treatment attenuates the clinical symptoms of RA patients and therefore has a therapeutic potential against RA [52,54].

### 2.6. Paeonia lactiflora Pallas

*Paeonia lactiflora* Pallas is a traditional Oriental natural plant medicine that has been used for thousands of years in China for its analgesic, anti-inflammatory, and immune system-improving efficacies. The therapeutic effects of *P. lactiflora* have been recognized by the Chinese Experience Medicine book, “Treatise on Cold Pathogenic” and “Synopsis of Golden Chamber” [55,56]. Total glycoside of paeony (TGP) is extracted from the root of *P. lactiflora*. TGP contains beneficial components, such as paeoniflorin, hydroxy-paeoniflorin, paeonin, albiflorin, and benzoyl-paeoniflorin (Figure 4) [57]. The first clinical trial of TGP was conducted in 1993 with 450 RA patients [58]. Therapeutic response was achieved in 71.7% of TGP-treated patients. Following the clinical trial of TGP for RA patients, a phase III trial was conducted with 1016 patients [59]. Based on these clinical trials, TGP was approved by the State Food and Drug Administration of China to enter the market as a disease-modifying drug for RA in 1998 [58]. Furthermore, a combined treatment with TGP and methotrexate showed a favorable effect on RA, with less side-effect [57]. TGP-treated RA patients showed decreased erythrocyte sedimentation rate and C-reactive protein level, along with a decrease in the population of IFN-γ- and IL-17-producing cells [60,61].

Paeoniflorin, a monoterpene glucoside, is a major active component of TGP, constituting over or equal to 40% of the total components. Paeoniflorin has been reported to possess extensive anti-inflammatory and immunoregulatory effects [57,62]. Paeoniflorin can diminish pain, joint swelling, synovial hypertrophy, bone erosion, and cartilage degradation in experimental arthritis [63,64]. It has been reported that paeoniflorin alleviated AA in rats by inhibiting DC maturation and activation [65]. Paeoniflorin also regulates immune function by affecting splenic T cells in rats with AA [66]. Clinical trials of paeoniflorin in the treatment of RA have been conducted in many hospitals in China. For example, paeoniflorin was shown to be a safer option to substitute DMARDs for long-term RA treatment [67]. As a result, paeoniflorin was approved for RA treatment and marketing in 1998 by the China Food and Drug Administration [68,69]. Clinical data have shown that paeoniflorin effectively relieves the symptoms and signs of RA without causing significant adverse effects [58].

In one study, 92 children with juvenile idiopathic arthritis hospitalized at Zhengzhou Children’s Hospital from March 2017 to March 2019 were treated with paeoniflorin. They were randomly divided into treatment and control groups (n = 46). The levels of IL-6, IL-1, and TNF-α in both groups were significantly lower after treatment than before treatment. The levels of IL-6, IL-1, and TNF-α in the treatment group were significantly lower than those in the control group [70]. Paeoniflorin is also recommended for the treatment of other autoimmune diseases, such as systemic lupus erythematosus, psoriasis, diabetes mellitus, diabetic nephropathy, ankylosing spondylitis, and immune liver injury [71,72,73,74].

### 2.7. Astragalus membranaceus Bunge

Radix Astragali (*Astragalus membranaceus* Bunge) is one of the most famous Oriental traditional medicines that has been used for many years [75]. It is sold worldwide as dietary supplements in the form of tea, beverages, soup, trail mix, and capsule [75,76]. Radix Astragali has been reported to exert hepatoprotective, antioxidative, antiviral, anti-hypertensive, and immunostimulatory properties [77,78]. It has also been reported to strengthen superficial resistance, drainage action, and new tissue growth [79,80]. Total flavonoids of astragalus (TFA) are the main active components isolated from A. membranaceus [77]. Jinxia et al. (2018) showed the immunomodulatory and anti-inflammatory mechanisms of TFA [81]. The mRNA expression levels of TNF-α, IL-6, IL-1β, IL-10, iNOS, and COX-2 were examined by RT-PCR in LPS-stimulated RAW 264.7 macrophages after treatment with TFA. The protein expression levels of iNOS, COX-2, MAPK, and nuclear factor (NF-kB) in LPS-stimulated RAW 264.7 macrophages were measured by western blotting. The results showed that TFA significantly decreased TNF-α, IL-1β, IL-6, iNOS, and COX-2 mRNA levels and increased IL-10 mRNA levels in LPS-stimulated RAW 264.7 cells in a dose-dependent manner [81]. Further studies revealed that TFA significantly inhibited the protein expression of iNOS and COX-2 as well as the phosphorylation of MAPKs (p38 and JNK) and NF-κB (IKKα/β, IκBα, NF-κB p65) in LPS-stimulated RAW 264.7 cells [81]. These results suggest that TFA exerts immunomodulatory and anti-inflammatory effects by regulating the MAPK and NF-κB signaling pathways in RAW 264.7 macrophages. 

TFA significantly inhibited serum TNF-α, IL-1β, PGE2, and the receptor activator of nuclear factor-κB ligand (RANKL) production [82]. Serum osteo-protegerin (OPG) production and OPG/RANKL ratio in rats were induced by Freund’s complete adjuvant (FCA) [82]. Histopathological examination indicated that TFA significantly attenuated inflammatory cell infiltration, synovial hyperplasia, pannus formation, and bone/cartilage damage [82]. In addition, the immunohistochemical assay showed that TFA inhibited NF-κB p65 expression in the synovial tissues of rats induced by FCA [82]. 

### 2.8. Achyranthes bidentata Blume 

*Achyranthes bidentata* Blume (ABB) is a species of flowering plant in the amaranth family. Amaranthaceae plants are traditional Oriental medicines that contain polysaccharides, triterpene saponins, sterones, and other active ingredients. The main pharmacological effects of ABB are antitumor, antiviral, anti-inflammatory, and analgesic [83]. ABB also exerts protective effect on rabbit knee joint cartilage and can inhibit cytokine IL-1β expression, increase TGF-β1 expression, and alleviate cartilage degeneration [84]. ABB treatment significantly reduced paw swelling, inflammatory cell proliferation, and bone degradation in CIA model rats [85]. The levels of fibrinogen, procollagen, protein disulfide isomerase A3, and apolipoprotein A-I were elevated in inflamed synovial tissue; however, the RA phenotypes were significantly reduced by ABB treatment [85]. In addition, α-1-antiprotease and manganese superoxide dismutase levels were increased in ABB-treated synovial tissues [85].

## 3. Treatment of RA with Mixed Herbal Compound

In recent years, researchers have confirmed that mixed herbal compound decoction can control RA by strengthening or inhibiting the production of anti-inflammatory factors, and regulating the immune and endocrine systems.

### 3.1. Wutou Decoction

Wutou decoction originates from “The Synopsis of the Golden Chamber” and is composed of ephedra, peony, astragalus, licorice, and Sichuan aconite. It has the functions of dispelling cold and dampness, removing numbness, and relieving pain in knee osteoarthritis [86]. A study found that Wutou decoction can inhibit synovial inflammation in knee osteoarthritis by regulating the TLR4/NF-κB signaling pathway [86]. In the study, using the random number table method, Wutou decoction was shown to effectively inhibit the expression of iNOS, TNF-α, and IL-6 [86]. Moreover, real-time PCR results showed that Wutou decoction inhibited TLR4, MyD88, TRAF6, and NF-κB p65 mRNA expression [86]. Western blotting results were consistent with real-time PCR results, in which Wutou decoction inhibited TLR4, MyD88, TRAF6, and NF-κB p65 protein expression [86].

### 3.2. GuiZhi ShaoYao ZhiMu Decoction

Tian et al. confirmed that GuiZhi ShaoYao ZhiMu Decoction (GSZD) regulates synovial cells [87]. GSZD, which also originates from “The Synopsis of the Golden Chamber,” is composed of Ramulus Cinnamomi, *P. lactiflora* root, Radix Glycyrrhizae Preparata, *Ephedra* sp., *Anemarrhena asphodeloides* Bunge root, *Atractylodes macrocephala*, and *Zingiber officinale*. In CIA model rats (AA models induced by acetic acid-soluble type II collagen and Freund’s complete adjuvant), GSZD increased Fas antigen expression and decreased Bcl-2 and p53 protein expression [87]. GSZD accelerated the clearance of synovial cells and significantly reduced synovial proliferation pathology in the CIA model [87]. GSZD attenuates RA by reversing the inflammation–immune system imbalance [88]. Combination treatment with GSZD and methotrexate was more efficacious and safer than methotrexate alone for treating RA [89]. This conclusion was based on 14 randomized controlled trials with 1224 RA patients [89]. Moreover, it has been reported that the effectiveness and safety of GSZD in treating RA are equal or superior to those of Western RA drugs [90].

## 4. Conclusions

Specific research and clinical data are available on the use of natural plant extracts or mixed herbal compounds (NPEMHCs) as a treatment of RA. From the pharmacological point of view, NPEMHCs effectively regulate the immune system to alleviate RA by inhibiting mainly pro-inflammatory cytokines. In addition, NPEMHCs attenuate the potential side effects of currently available drugs. Further, individualized medication can be applied according to the physical condition of each individual patients. However, thus far, the pathogenesis of RA and its immune mechanism has not been fully understood. Therefore, the treatment of RA by NPEMHC needs more basic research support, and a clinical study involving a large group of patients will help advance the development of NPEMHCs as drugs for RA treatment. 

## Figures and Tables

**Figure 1 medicina-57-00266-f001:**
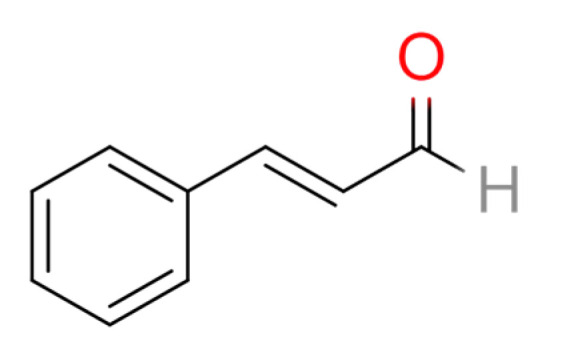
Chemical structure of cinnamaldehyde.

**Figure 2 medicina-57-00266-f002:**
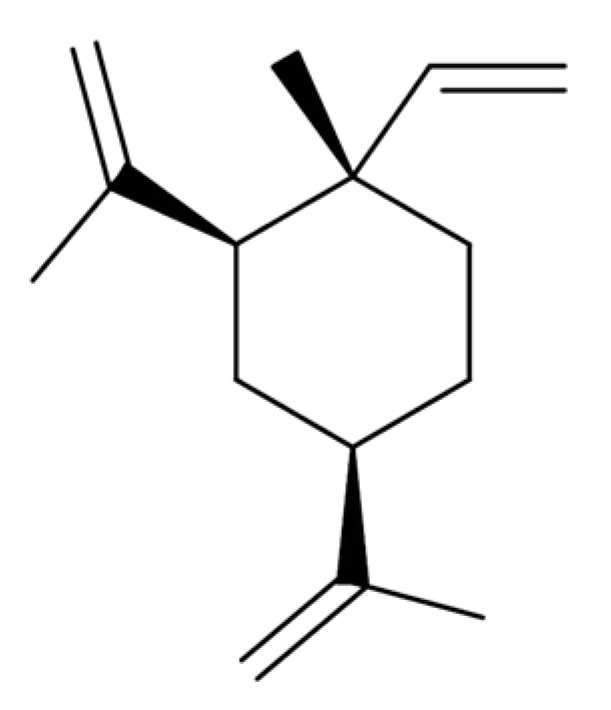
Chemical structure of β-elemene.

**Figure 3 medicina-57-00266-f003:**
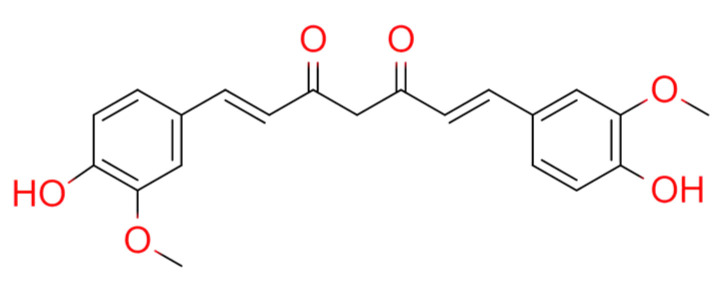
Chemical structures of curcumin.

**Figure 4 medicina-57-00266-f004:**
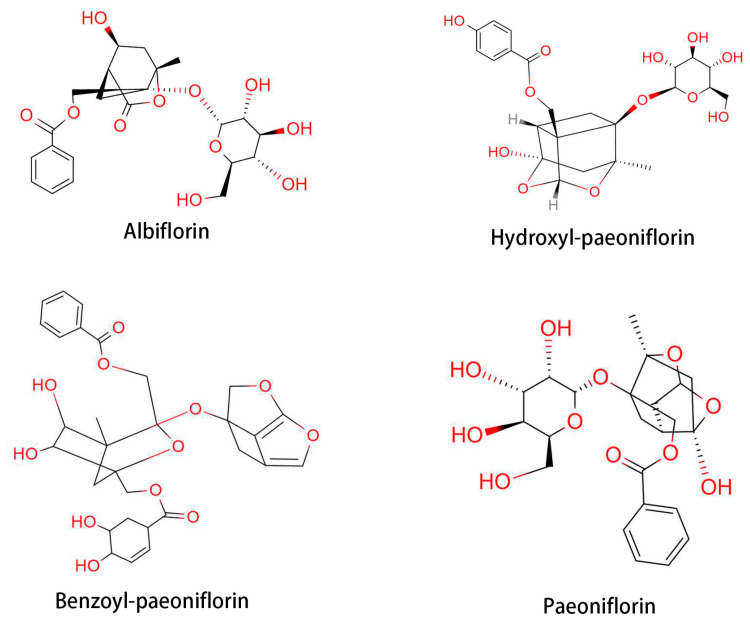
Structures of the principal constituents in total glycoside of paeony (TGP).

## Data Availability

Data available in a publicly accessible repository.

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
