# Peer review of "Natural Plant Extracts and Compounds for Rheumatoid Arthritis Therapy"

_medicina, 2021, doi:10.3390/medicina57030266_

Round 1

Reviewer 1 Report

This paper summarises the current evidence on natural plant extracts and their use to treat rheumatoid arthritis.

Major comments

Much of the evidence is extrapolated from in vitro studies or studies in laboratory animals with collagen induced arthritis (CIA). The authors do not always make this clear. In the abstract they say ‘natural plant extracts….effectively regulate the immune system to alleviate RA..’ but there is no direct evidence to support this statement in the paper. Throughout the paper the authors have over-stated the likely benefits of the compounds and imply the limited in vitro and animal studies show that the compounds are effective in human RA. They need to make it clear that although these compounds may have biological effects the current evidence that they ameliorate human RA is limited.

The authors describe Rheumatoid arthritis (RA) as an autoimmune disease of the joint but it is a systemic disease which affects multiple organ systems not just the joints. It causes inflammation within the joints which can affect activity and function, but effective early treatment can prevent joint damage and preserve function.

The statement ‘..clinical trials have proved that NPEC can alleviate RA..’ is misleading and factually incorrect, They refer to a paper (Xu et al 2015) which examined adjuvant induced arthritis in rats.

The statement ‘..CA …can inhibit RA progression..’ overstates the evidence. Liu et al 2020 demonstrated amelioration of synovial inflammation in adjuvant arthritis in rats but did not claim to have inhibited RA progression.

Reference 15 describes Autosomal recessive hyponatremia due to isolated salt wasting in sweat associated with a mutation in the active site of Carbonic Anhydrase 12 and does not discuss the benefits of L.chuangxiong as claimed in the paper.

‘wind-dampness’ is not a recognised medical term or condition.

Minor comments

The abbreviation CIA is not explained (I assume it refers to collagen induced arthritis)

‘MHC’ usually refers to the Major Histocompatability Complex . It’s use as an abbreviation for Mixed Herbal Compound is confusing in a paper aimed at a medical readership

Author Response

Major comments

Comment 1-1.

Much of the evidence is extrapolated from in vitro studies or studies in laboratory animals with collagen induced arthritis (CIA).

[Response]

We have added details regarding the adjuvant-induced arthritis model and human clinical trials in the text (see comments1-2 and 1-4).

Comment 1-2.

The authors do not always make this clear. In the abstract they say ‘natural plant extracts….effectively regulate the immune system to alleviate RA..’ but there is no direct evidence to support this statement in the paper.

[Response]

We have added details about the immunological effects of the natural plant extracts in the text.

  • The last part of “2.4. Tripterygium wilfordii Hook F” (page 4)
  • The last part of “2.5. Curcumae Longae Rhizoma” (page 5)
  • The first part of “2.6. Paeonia lactiflora Pallas” (page 5)
  • The middle part of “2.6. Paeonia lactiflora Pallas” for paeoniflorin (page 6)
  • The last part of “3.2. GuiZhi ShaoYao ZhiMu Decoction” (page 8)

Comment 1-3.

 Throughout the paper the authors have over-stated the likely benefits of the compounds and imply the limited in vitro and animal studies show that the compounds are effective in human RA.

[Response]

We have eliminated such over-statements in the text. Instead, we have added information based on clinical trials on RA treatment using natural compounds. 

Comment 1-4.

They need to make it clear that although these compounds may have biological effects the current evidence that they ameliorate human RA is limited.

[Response]

We have added more information about treatment of RA with natural compounds.

  • page 2 for cinnamaldehyde
  • page 3 for ligustrazine
  • page 5 for curcumin
  • page 5 for total glycoside of paeony (TGP)
  • page 8 for GuiZhi ShaoYao ZhiMu Decoction (TGP)

Comment 2.

The authors describe Rheumatoid arthritis (RA) as an autoimmune disease of the joint but it is a systemic disease which affects multiple organ systems not just the joints. It causes inflammation within the joints which can affect activity and function, but effective early treatment can prevent joint damage and preserve function.

[Response]

As per the reviewer’s comments, we have added details about RA as an autoimmune disorder in the text (first part of the introduction, page 1).

Comment 3.

The statement ‘..clinical trials have proved that NPEC can alleviate RA..’ is misleading and factually incorrect, They refer to a paper (Xu et al 2015) which examined adjuvant induced arthritis in rats.

[Response]

We have revised “clinical trials” to “preclinical trials”. Note that the “adjuvant-induced arthritis model” is one of the most popular rheumatoid arthritis animal models.

Comment 4.

The statement ‘..CA …can inhibit RA progression..’ overstates the evidence. Liu et al 2020 demonstrated amelioration of synovial inflammation in adjuvant arthritis in rats but did not claim to have inhibited RA progression.

[Response]

We have provided this statement as the study’s conclusion. We have included the following sentence: “The researchers concluded that CA is a potential therapeutic compound that can inhibit RA progression by suppressing IL-1β, modulating the succinate/HIF-1α axis, and inhibiting NLRP3 [12]”

Comment 5.

Reference 15 describes Autosomal recessive hyponatremia due to isolated salt wasting in sweat associated with a mutation in the active site of Carbonic Anhydrase 12 and does not discuss the benefits of L.chuangxiong as claimed in the paper.

[Response]

We have replaced the citations with the following preferred references.

  1. Yuan, X., et al., Chemical constituents of Ligusticum chuanxiong and their anti-inflammation and hepatoprotective activities. Bioorg Chem, 2020. 101: p. 104016.
  2. Du, J.C., et al., [Research progress of chemical constituents and pharmacological activities of essential oil of Ligusticum chuanxiong]. Zhongguo Zhong Yao Za Zhi, 2016. 41(23): p. 4328-4333.
  3. Huang, C., et al., A pectic polysaccharide from Ligusticum chuanxiong promotes intestine antioxidant defense in aged mice. Carbohydr Polym, 2017. 174: p. 915-922.

Comment 6.

‘wind-dampness’ is not a recognised medical term or condition.

[Response]

We have replaced “wind-dampness” with “painful obstruction syndrome”

Minor comments

Comment 7.

The abbreviation CIA is not explained (I assume it refers to collagen induced arthritis)

[Response]

We apologize for this oversight. We have defined the abbreviation “collagen-induced arthritis (CIA)” at the first instance.

Comment 8.

‘MHC’ usually refers to the Major Histocompatability Complex . It’s use as an abbreviation for Mixed Herbal Compound is confusing in a paper aimed at a medical readership

[Response]

We agree with your comment. We have deleted the MHC abbreviation throughout the document.

Reviewer 2 Report

In this manuscript, the author reviewed the specific research and clinical data on using natural plant extracts or mixed herbal compounds (NPEMHCs) as a treatment of rheumatoid arthritis (RA). They pointed out that the pathogenesis of RA and its immune mechanism has not been fully understood. Thus, the treatment of RA by NPEMHC needs more basic research support, and a clinical study involving a large group of patients will help advance the development of NPEMHCs. This work's subject matter is interesting; however, much improvement can still be made to make it more straightforward and more concise. 1. As mentioned in the introduction, advances in the treatment of RA have been remarkable. The authors should also mention the relationship of NPECs to methotrexate and biologic agents used for RA. 2. Although in vitro results of NPECs have been shown, the clinical significance of NPECs has not been clarified. The authors should indicate what they consider to be the place of NPECs in the treatment of RA. 3. Please mention the cautions for clinical application of NPECs, e.g., side effects.

Author Response

Comments and Suggestions for Authors (Reviewer 2)

In this manuscript, the author reviewed the specific research and clinical data on using natural plant extracts or mixed herbal compounds (NPEMHCs) as a treatment of rheumatoid arthritis (RA). They pointed out that the pathogenesis of RA and its immune mechanism has not been fully understood. Thus, the treatment of RA by NPEMHC needs more basic research support, and a clinical study involving a large group of patients will help advance the development of NPEMHCs. This work's subject matter is interesting; however, much improvement can still be made to make it more straightforward and more concise.

Comment 1.

  1. As mentioned in the introduction, advances in the treatment of RA have been remarkable. The authors should also mention the relationship of NPECs to methotrexate and biologic agents used for RA.

[Response]

We have discussed about the combination treatment of NPECs plus methotrexate. For example, total glycoside of paeony (TGP) with methotrexate on page 3 and GuiZhi ShaoYao ZhiMu Decoction (GSZD) with methotrexate on page 8.

Comment 2.

  1. Although in vitro results of NPECs have been shown, the clinical significance of NPECs has not been clarified. The authors should indicate what they consider to be the place of NPECs in the treatment of RA.

[Response]

In line with the reviewer’s comment, we have discussed the clinical significance of NPECS on page 2 for cinnamaldehyde; on page 3 for ligustrazine; on page 5 for curcumin; on page 5 for total glycoside of paeony (TGP); and on page 8 for the GuiZhi ShaoYao ZhiMu Decoction (TGP)

Comment 3.

  1. Please mention the cautions for clinical application of NPECs, e.g., side effects.

[Response]

Long-term use of currently available drugs for the treatment of RA can cause side effects. Natural plant extracts and compounds were developed to overcome this issue. No side effects have been reported with the use of the developed natural compounds for the RA drug. 

Round 2

Reviewer 1 Report

The authors have addressed the comments and concerns raised in my reivew